# *Sitophilus zeamais* (Coleoptera: Curculionidae) Activity and Germination of Corn Seeds Stored under Vacuum Pressure

**Arturo Mancera-Rico [1,*], Mario E. Vázquez-Badillo [1], Ernesto Cerna-Chávez [2], Miriam Sánchez-Vega [3] and Elizabeth González-Estrada [4]**

1   Centro de Capacitación y Desarrollo en Tecnología de Semillas, Universidad Autónoma Agraria Antonio Narro, Saltillo 25315, Mexico; marioe.vazquez@hotmail.com
2   Parasitolgía Agrícola, Universidad Autónoma Agraria Antonio Narro, Saltillo 25315, Mexico; jabaly1@yahoo.com
3   Consejo Nacional de Humanidades, Ciencia y Tecnología, Mexico City 03940, Mexico; sanvemi16@gmail.com
4   Socioeconomics, Statistics, and Informatics, Colegio de Postgraduados, Texcoco 56220, Mexico; egonzalez@colpos.mx
*   Correspondence: mncrarico@gmail.com

**Abstract:** Warehouse pests cause losses in seed quality by physically damaging seeds, carrying other pathogens, and leaving residues of their consumption or their remains. Taking into account that warehouse pests have higher metabolism rates than seeds, in the present study, the viability of *Sitophilus zeamais* (Motschulsky & V.de, 1855) insects and the germination of corn seeds were evaluated for 93 and 180 days, respectively, under conditions of a partial vacuum (0.26 atm (atmosphere)), hermetic sealing, and air exchange. The partial vacuum environment (0.26 atm) did not negatively affect the germination of the seeds over a period of 180 days and resulted in the mortality of adult insects from the 4th day onwards; it also completely prevented physical damage to the seeds. The hermetic packaging did not negatively affect germination, but the insects remained active until day 34, and there was severe damage to the seeds used for their sustenance. Similarly, the gas exchange packaging did not affect germination, but the insects remained active until the last day of observation (93) and also caused severe damage to the seeds used for their maintenance. Partial vacuum (0.26 atm) storage represents a practical solution in certain conditions, for example, the short-term (6 months) storage of germplasms at room temperature (24–26 °C); it can also be useful in the storage and transfer of grains, with the possibility of eliminating or reducing the need for insecticide applications.

**Keywords:** maize weevil; germplasm; storage; oxygen; grain

## 1. Introduction

Warehouse pests cause losses in seed quality by physically damaging seeds, carrying other pathogens, and leaving residues of their consumption or their remains. The maize weevil, *Sitophilus zeamais* (Motschulsky & V.de, 1855), is the most significant pest for stored corn seed and grain. In order to avoid and control its attacks, it is necessary to apply synthetic and natural products, some of which are harmful to the applicator and surrounding living organisms [1–3], and contamination has been reported in water sources such as rivers, lakes, and underground aquifers. Due to the development of resistance by insects, an integrated pest management system is recommended, and to facilitate this, a diversification of the means of control is needed.

Seeds are susceptible to damage by insecticides and fungicides [4–6]; thus, when employed, it is important to carefully control the exposure period and dose. Furthermore, when used on edible grains, intoxication by the consumer must be avoided. Reducing insecticide applications would reduce risks to applicators, consumers, wildlife, air, and water sources. Preserving seeds without synthetic products may represent competitive

advantages, especially for organic production systems, and seeds can simply be stored free of insecticides or fungicides for any other requirement.

To prevent insect and fungal attacks on seeds and grains, some researchers have studied the benefits of atmospheres saturated with gases such as ozone, nitrogen, and carbon dioxide [7–9]. Vacuum pressure is another alternative for preserving seeds and grains, and some authors have studied its utility for controlling insects on grains [10], but its application for seed preservation is not well documented yet, despite the important advancements achieved by authors like Lawrence, Bicksler, Duncan [11], and Prasantha [12].

Seeds are living organisms with reduced metabolic activities as compared to insects and other organisms; it is possible that seeds can be preserved under conditions not favorable to insect and mold growth and proliferation. Seeds store well with moisture contents as low as 1.5% (w.b. (wet basis)) [13], temperatures as low as $-196\ °C$ [14], and low oxygen contents. The respiration rate of pea (*Cicer arietinum* L.) seeds at a moisture content of 60% (w. b.) is 240 $\mu$mol $O_2$ $h^{-1}$ $g^{-1}$ dry matter [15], while the respiration rate of barley (*Hordeum vulgare* L.) seeds at 20% (w.b.) and 25 $°C$ is 2 $\mu$L $O_2$ $h^{-1}$ $g^{-1}$ dry matter, increasing 100 times at 30% (w.b.), which is linearly related to the moisture content [16]. Oxygen is essential to germination, as respiration is prevented or delayed when oxygen is not fully available [17]. Vacuum pressure is related to the improvement of vigor and longevity in primed sh-2 sweet corn seeds [18], which are represented by the emergence percentage and seedling dry weight, respectively. Deepa and Chetti [19] found that chili (*Capsicum annuum* L.) seeds store well under vacuum pressure, as reflected in biochemical parameters such as capsaicin content (0.06–0.07 vs. 0.02–0.03% under nonvacuum), protein content (69.6–71.2 vs. 42.4–60.8 mg/g under nonvacuum), and carbohydrate content (260.0–370.0 vs. 120.0–140.0 mg/g under nonvacuum). Lawrence, Bicksler, and Duncan [8] found that, after 1 year of vacuum storage, Lablab (*Lablab purpureus* L.) seed viability was 77.6%, compared with 66.5% under nonvacuum conditions.

On the other hand, for weevil development, the optimum temperature ranges from 20 to 30 $°C$ [20,21]. Relative humidity should be about 70% to keep them well hydrated and able to carry out their normal activities [21–23]. Oxygen consumption under normal conditions is about 0.04 $\mu$mol $h^{-1}$ $mg^{-1}$ (of living weight) for *Tribolium castaneum* (Herbst, 1797) and *Tenebrio molitor* (Linnaeus, 1758), and as low as 0.005 and 0.006 for *Cryptoglossa muricata* (LeConte, 1851) and *Eleodes longicollis* (LeConte, 1851), respectively [24]. Eggs and adults of *S. zeamais* are more susceptible to anoxia than larvae and pupae [25]. Atmospheric pressure is relevant for insects' breathing. Insects exchange gas in a cyclic and discrete way [26], aided by pressure gradients [27]; thus, vacuum pressure might interfere with the natural gradients and restrict breathing capability. The drugstore beetle (*Stegobium paniceum* Linnaeus, 1758), southern cowpea weevil (*Callosobruchus chinensis* Linnaeus, 1758), and red flour beetle (*Tribolium castaneum* Herbst, 1797) exposed to low-pressure conditions (20 and 10 kPa (kilo Pascal)) showed mortality percentages related to seed species and exposure time (8 and 32 h) [9]. The cowpea bruchid (*Callosobruchus maculatus* Fabricius, 1775) population was lower (4.9 insects per 50 seeds) when exposed to low pressure (80 kPa) compared with nonvacuum conditions (123.3 insects per 50 seeds) [8]. It is expected that 100% mortality can be achieved by using exposure times longer than those studied by Prasantha [9] (32 h) and pressure lower than that studied by Lawrence, Bicksler, and Duncan [8] (80 kPa). Variables such as species of insects and seeds, temperature, and relative humidity must be taken into account. Despite their apparent susceptibility to stress factors, insects may survive unfavorable ambient conditions by means of diapause and metabolic arrest [28,29]. *Manduca sexta* (Linnaeus, 1763) pupae and larvae recovered from drowning for 5 and 4 days, respectively [30]. At vacuum pressure, loss of water might occur [31], which must be taken into account to preserve seeds; seeds usually store well at lower moisture contents, but insects may dehydrate.

The objective of this study was to identify and evaluate a method for the control of corn weevil in seeds and stored grains that does not present a toxic danger to the applicator,

the consumer, wildlife, or the environment. According to the obtained results, the partial vacuum environment can be used to safely store seeds and grains free of any toxic products.

## 2. Materials and Methods

Seeds, without chemical treatment, of the La Gloria HAN-421 corn hybrid produced two years before the study were used. The insects were obtained from a grain warehouse and placed in 946 mL flasks with gas exchange and sufficient corn grain (50% of the volume) for their reproduction. Adult insects were left for 15 days and later extracted. The reproduction chamber was monitored weekly, and after 60 days, the adult weevils were extracted from them and used in the study. The insects were roughly 60–75 days old at the beginning of the tests. The temperature of the chambers was kept at 26 ± 0.5 °C during the day (12 h) and 24 ± 0.5 °C at night (12 h) during the experiment and reproduction, for which an environmental chamber (Biotronette Mark III, Labline Instruments, Melrose Park, IL, USA) was used.

The storage chambers used in the evaluation consisted of 946 mL glass bottles of the brand Mason Ball model S-17492 (Hearthmark, LLC, dba Jarden Home Brands, Fishers, IN, USA) with a regular mouth with a 2 3/8 in [inch] internal opening and an airtight tin lid. These bottles comply with the FDA (Food and Drug Administration), with a height and width of 17 × 9 cm. The air exchange chambers were closed with filter paper that allowed gas exchange and prevented the entry or exit of insects. The sealed chambers were closed with a hermetic lid that prevented gas exchange. The vacuum chambers were closed with a vacuum closure lid, which was drilled to insert and fix a vacuum valve (valve for air conditioning and refrigeration); to bond the valve to the lid, leak-proof epoxy glue Devcon 5 Minute Epoxy (ITW Performance Polymers, Danvers, MA, USA) was used. To create the vacuum, an FJC 9281 (FJC, Mooresville, NC, USA) vacuum pump equipped with pressure gauges and pressure regulators was used; the vacuum pressure was checked and adjusted by means of weight differences by using a scale (AND GF-2000, San Jose, CA, USA).

### 2.1. Studied Independent Variables

Atmospheric pressure and gas exchange. Three storage conditions were evaluated: 0.26 ± 0.02 atm (atmosphere) in the vacuum chamber, 1 atm in the sealed chamber, and 1 atm in the air exchange chamber. The vacuum was first measured with manometers; however, to reduce the error caused by gas exchange when removing the hose, it was adjusted by weighing. Under the conditions of the experiment, each chamber contained 1.176–1.194 g of air (since its net volume was 953–968 mL), and 72 to 76% of the air (0.86–0.91 g of air) was extracted to achieve an average pressure of 0.26 atm (0.28 to 0.24 atm); this weight was monitored weekly during the experiment, although the chambers used were subjected to a prior evaluation for 30 days to ensure their tightness. The weight of the open vacuum chambers together with their lids, without samples, was 407–414 g.

Storage period. Insect viability was evaluated daily for 93 days, and seed germination was evaluated at 0, 30, 150, and 180 days. In both cases, the same environmental conditions were maintained, with a temperature of 26 ± 0.5 °C during the day (12 h) and 24 ± 0.5 °C at night. Taking into account that seeds and grains can be stored for up to several months or years, it was decided to evaluate the germination of the seeds in regular periods, up to 180 days. On the other hand, we wanted to evaluate whether the vacuum chamber allowed for the eradication of the insects studied within a period similar to that required in fumigation, which is usually 4 to 5 days, as well as the probable growth of the population in the three types of chambers, in which a new generation could be presented after 30–35 days (new births were not included in the analysis). We also wanted to rule out the effect of the natural aging of the insects on their deaths, for which the number of live insects was evaluated daily for a period of 93 days.

*2.2. Experimental Design*

Two experiments were conducted. In the first experiment, the germination of the seeds was evaluated as a function of the type of chamber (air exchange, sealed, and vacuum) and storage period (0, 30, 150, and 180 days). For this, a completely randomized factorial design with two study factors was used. Each experimental unit consisted of a chamber with 50 corn seeds without insects, and three repetitions per treatment were used. In the second experiment, a completely randomized factorial design was used to evaluate the viability of the insects. Each experimental unit consisted of 15 insects and 15 insecticide-free corn grains for their sustenance, six repetitions per treatment were used. Each treatment consisted of a combination of the two studied factors: type of chamber (air exchange, sealed, and vacuum) and storage period (the viability of the insects was determined daily for 93 days). In addition, estimates of the availability of air and oxygen per insect and the air renewal rate for the air exchange chamber were made, and comparisons of the availability of air and oxygen between the different types of chambers were carried out using a completely randomized factorial design, in which the type of chamber was the studied factor. At the same time, a completely randomized factorial design was used to assess the damage inflicted by the insects on the seed corns; the damage was evaluated at the end of the 93 days for each type of chamber (air exchange, sealed, and vacuum).

*2.3. Evaluated Dependent Variables*

Standard germination was determined using the "between paper" methodology recommended by ISTA [32]; paper rolls were hydrated with distilled water, the chamber temperature was maintained at 24–28 °C, and the germination count was carried out on the seventh day. Only normal seedlings were counted as having germinated. The aim was to identify if there was a negative or positive effect on germination caused by the vacuum chamber and storage period.

Insect viability (alive insects) was determined by observing insect motility and posture. Nonmoving insects with tightly curled legs were determined to be alive because they were later found crawling inside the chamber; in contrast, nonmoving insects with loose legs or somewhat straighter, less curved legs than described previously were determined to be dead, as they remained that way thereafter. New births were not counted. The aim was to identify if there was a negative or positive effect on insect viability caused by the type of chamber and storage duration.

Grains damaged. In the corn grains used as sustenance for the insects, the level of damage caused by the insects was determined on a visual scale from 0 to 5, where 5 corresponded to the maximum level of damage and 0 to zero damage. To evaluate the damage, the presence of perforated grains and loose dust was taken into account. The aim was to evaluate the damage caused by insects at the end of the storage period in the three storage conditions: vacuum, hermetic closure, and gas exchange.

The availability of air and oxygen per insect was estimated using Equations (2) and (3), which consider the sum of the daily counts of live insects (Equation (1)) and the air volume in the vacuum (pressure factor *PF* equals 0.26) and sealed chambers (*PF* equals 1.00).

$$Insects * day = \sum_{i=1}^{n}(insect\ count\ in\ day\ i) \tag{1}$$

$$Daily\ air\ availability\ per\ insect = \frac{Chamber\ volume\ (mL)(PF)}{Insects * day} \tag{2}$$

$$Daily\ oxygen\ availability\ per\ insect = 0.21 * \frac{Chamber\ volume\ (mL)(PF)}{Insects * day} \tag{3}$$

The purpose of estimating the availability of air and oxygen per insect was to identify the factor with the highest influence on the death of insects: oxygen depletion or alteration in the respiration rate due to atmospheric depression.

The air renewal rate for the air exchange chamber, occasioned by air volume changes due to temperature fluctuations, was estimated by means of the ideal gas equation (Equation (4)). For the calculation, it was assumed that the air mass remains constant, as it is, from its initial state ($T_1$ and $V_1$) to the final state ($T_2$ and $V_2$) in each cycle of expansion or compression; for this reason, Equation (5) was derived from Equation (4).

$$PV = mRT \tag{4}$$

If the mass and pressure remain constant in a process of temperature change,

$$\frac{V_1}{T_1} = \frac{V_2}{T_2} \tag{5}$$

The purpose of estimating the air renewal rate in the air exchange chamber was to determine the number of days for total air renewal under ideal conditions, as well as to identify if the air renewal is sufficient to avoid a decrease in the amount of oxygen, which could negatively affect the viability of the insects.

### 2.4. Data Analysis

For data analysis, an analysis of variance and Tukey comparisons of means were performed. Analysis of variance and Tukey comparisons of means were used for two reasons: the first is that they are sufficient for comparing the effects caused by the applied treatments, and the second is that they are relatively simple statistical tests and widely known in the scientific community.

Data in percentages and proportions do not facilitate the condition of normality in performing analysis of variance. To correct this, the original data, namely the percentages and damage index, were transformed with the arcsine $(Y/100)^{1/2}$ and arcsine $(Y/5)^{1/2}$ functions, where Y is the original data. After that, analysis of variance was performed, and, in the case of meaningful effects, Tukey comparisons of means were carried out. The comparisons of means are shown with the transformed data to avoid rounding or calculation errors when retransforming the value that allows comparisons to be made (LSD). To estimate the percentages and damage index equivalent to the original data, it was necessary to retransform them, applying the inverse function. Analysis of variance and Tukey comparisons of means were performed with the aid of SAS® Studio [33].

In the first experiment, to compare the germination percentages of the seeds according to the type of chamber and the storage period, an analysis of variance was carried out. Before the analysis of variance, the germination percentages (Y) were transformed (T) with an arcsine function [$T = \text{arcsine } (Y/100)^{1/2}$]. In the second experiment, to compare the percentages of live insects depending on the type of chamber and the storage period, an analysis of variance and Tukey comparisons of means were performed. Before the analysis of variance, the percentages of live insects were transformed with the arcsine function. Additionally, in the second experiment, to estimate the volume of air renewal in the chamber with gas exchange caused by day/night temperature changes, the ideal gas equation was used; then, to evaluate the daily availability of air and oxygen in each chamber, an analysis of variance and Tukey comparisons of means were performed. Additionally, in the second experiment, to evaluate the damage caused by insects, depending on the type of chamber and the storage period, an analysis of variance and Tukey comparisons of means were performed, for which the original and transformed data were used, where the damage index (Y) was transformed (T) with an arcsine function [$T = \text{arcsine } (Y/5)^{1/2}$].

## 3. Results

### 3.1. Standard Germination

There was no significant effect ($\alpha = 0.05$) caused by the type of chamber, storage period, or the interaction of the chamber and storage period on germination (analysis of variance not shown). The average germination percentages were 87.00, 89.00, and 87.33% for the

vacuum, sealed, and air exchange chambers, and 87.56, 88.00, 87.33, and 88.22% after 0, 30, 150, and 180 days of storage under those conditions (the averages were estimated with the transformed data).

### 3.2. Insect Viability

There was a significant effect ($\alpha = 0.05$) caused by the type of chamber, storage period, and the interaction of the chamber and storage period on weevil viability (analysis of variance not shown). The vacuum chamber was related to the lowest average percentage of weevil viability (Table 1). The mean percentage of weevil viability decreased over time (Table 2). In the vacuum and sealed chambers, there was no increase in the number of insects, while in the air exchange chamber, only two new adult insects were observed (not included in the analysis). Insect viability in the vacuum chamber was drastically reduced between day 0 and day 4 (Table 3), while in the sealed chamber, a reduction was observed between days 20 and 30; in the air exchange chamber, there was no noticeable reduction in the viability of the insects, since on day 93, it remained at 63.33%. The interaction between the type of chamber and the storage period was significant; this means that the mortality trend was different between the three types of chambers, which can be seen in Table 3, since it is evident that in the vacuum chamber, the chamber with hermetic closure, and the chamber with gas exchange, the mortality was fast, medium, and slow, respectively.

**Table 1.** Type of chamber and average percentage of weevil viability during 93 days of storage.

| Chamber | Alive Insects, % |
|---|---|
| Air exchange chamber (1 atm (atmosphere)) | 68.44 a |
| Sealed chamber (1 ± 0.08 atm) | 25.25 b |
| Vacuum chamber (0.26 ± 0.04 atm) | 2.18 c |
| HSD (Tukey, 0.05) | 1.3478 |

Means with the same letter are not statistically different. Pressure variations are due to the vacuum procedure (vacuum chambers) and temperature changes (vacuum and sealed chambers). The percentages of weevil viability (Y) used for the analysis were transformed (T) with an arcsine function [T = arcsine $(Y/100)^{1/2}$].

**Table 2.** Period of storage and average percentage of weevil viability in vacuum chambers (0.26 ± 0.04 atm (atmosphere)), hermetically sealed chambers (1 ± 0.08 atm), and air exchange chambers closed with filter paper (1 atm).

| Period, Days | Alive Insects, % | Tukey Groups | | | | | | | |
|---|---|---|---|---|---|---|---|---|---|
| 0 | 89.99 | a | | | | | | | |
| 1 | 79.44 | a | b | | | | | | |
| 2 | 71.22 | | b | c | | | | | |
| 3 | 66.71 | | b | c | d | | | | |
| 4 | 58.30 | | | c | d | e | | | |
| 5 | 58.30 | | | c | d | e | | | |
| 10 | 55.74 | | | | d | e | | | |
| 20 | 52.01 | | | | | e | f | | |
| 30 | 29.15 | | | | | | | h | i |
| 40 | 23.43 | | | | | | | | i |
| 50 | 22.39 | | | | | | | | i |
| 60 | 18.91 | | | | | | | | i |
| 70 | 18.47 | | | | | | | | i |
| 80 | 18.47 | | | | | | | | i |
| 93 | 18.47 | | | | | | | | i |
| HSD (Tukey, 0.05) | 13.785 | | | | | | | | |

Means with the same letter are not statistically different. Pressure variations are due to the vacuum procedure (vacuum chambers) and temperature changes (vacuum and sealed chambers). The percentages of weevil viability (Y) used for the analysis were transformed (T) with an arcsine function [T = arcsine $(Y/100)^{1/2}$].

**Table 3.** Average percentage of weevil viability in vacuum chambers ($0.26 \pm 0.04$ atm (atmosphere)), hermetically sealed chambers ($1 \pm 0.08$ atm), and air exchange chambers closed with filter paper (1 atm) during 93 days of storage.

| Time of Storage, Days | Alive Insects, % | | | | | |
|---|---|---|---|---|---|---|
| | Vacuum Chamber | | Sealed Chamber | | Air Exchange Chamber | |
| 0 | 89.99 | a | 89.99 | a | 89.99 | a |
| 1 | 58.31 | bcde | 89.99 | a | 89.99 | a |
| 2 | 33.67 | efg | 89.99 | a | 89.99 | a |
| 3 | 22.67 | fgh | 89.99 | a | 87.44 | a |
| 4 | 0.00 | h | 87.44 | a | 87.44 | a |
| 5 | 0.00 | h | 87.44 | a | 87.44 | a |
| 10 | 0.00 | h | 84.89 | a | 84.89 | a |
| 20 | 0.00 | h | 77.22 | abc | 78.81 | abc |
| 30 | 0.00 | h | 12.44 | gh | 75.01 | abc |
| 34 | 0.00 | h | 4.43 | h | 75.01 | abc |
| 35 | 0.00 | h | 0.00 | h | 75.01 | abc |
| 40 | 0.00 | h | 0.00 | h | 70.29 | abc |
| 50 | 0.00 | h | 0.00 | h | 67.18 | abc |
| 60 | 0.00 | h | 0.00 | h | 56.73 | bcde |
| 70 | 0.00 | h | 0.00 | h | 55.40 | cde |
| 80 | 0.00 | h | 0.00 | h | 55.40 | cde |
| 90 | 0.00 | h | 0.00 | h | 55.40 | cde |
| 93 | 0.00 | h | 0.00 | h | 55.40 | cde |
| HSD (Tukey, 0.05) | 26.435 | | | | | |

Only selected days and regular intervals of 10 are shown. Means with the same letter (across the whole table) are not statistically different. Pressure variations are due to the vacuum procedure (vacuum chambers) and temperature changes (vacuum and sealed chambers). The percentages of weevil viability (Y) used for the analysis were transformed (T) with an arcsine function [$T = \arcsin (Y/100)^{1/2}$].

### 3.3. Air and Oxygen Availability per Insect

The average daily air renewal rate in the air exchange chamber was 16% (Table 4); this renewal was attributed only to the air expansion and contraction provoked by the temperature range of the experiment (24–26 °C), without considering the gas diffusion itself. There was a significant effect ($\alpha = 0.05$) caused by the type of chamber on the availability of air, oxygen, and oxygen per mg of live weight (analysis of variance not shown); the availability of oxygen per unit mass of the insects was 0.2679, 0.9669, and 1.9643 $\mu$mol h$^{-1}$ mg$^{-1}$ in the sealed, vacuum, and gas exchange chambers, respectively (Table 5).

**Table 4.** Air renovation volume caused by changes in temperature in the air exchange chamber under ideal conditions.

| $V_1$, mL | $T_1$ °C | $T_2$, °C | $V_2$, mL $V_2 = V_1 \cdot T_2/T1$ | $\Delta V$, mL $\Delta V = V_2 - V_1$ | $\Delta Vu$, mL/mL $\Delta Vu = \Delta V/V_1$ |
|---|---|---|---|---|---|
| 946 | 26 | 24 | 873.23 | −72.77 | −0.08 |
| 946 | 24 | 26 | 1024.83 | 78.83 | 0.08 |
| TOTAL | --- | --- | --- | 151.60 | 0.16 |

Calculations were performed using the ideal gas equation.

### 3.4. Grain Damage

There was a significant effect ($\alpha = 0.05$) caused by the type of chamber on the damage index (analysis of variance not shown). On a scale from 0 to 5, the level of damage was 0.00, 4.17, and 4.83 in the vacuum, sealed, and air exchange chambers, respectively (Figure 1, Table 6). In the vacuum chamber, it was visually observed that the activity of the insects was altered from the moment the vacuum was applied, although they retained a certain level of activity until the fourth day; meanwhile, in the sealed and air exchange chambers, their activity was maintained for a longer time (Table 3).

**Table 5.** Daily availability of air and oxygen per insect in the vacuum (0.26 ± 0.04 atm (atmosphere)), sealed (1 ± 0.08 atm), and air exchange (1 atm) chambers.

| Chamber | Air Availability [mL Insect$^{-1}$ Day$^{-1}$] | | O$_2$ Availability [mL Insect$^{-1}$ Day$^{-1}$] | | O$_2$ Availability per mg of Live Weight [μmol h$^{-1}$ mg$^{-1}$] | |
|---|---|---|---|---|---|---|
| Air exchange | 16.731 | a | 3.514 | a | 1.9643 | a |
| Vacuum | 8.236 | b | 1.730 | b | 0.9669 | b |
| Sealed | 2.282 | c | 0.479 | c | 0.2679 | c |
| LSD (Tukey, 0.05) | 2.77 | | 0.58 | | 0.325 | |

Averages in columns with the same letter do not differ statistically. Pressure variations are due to the vacuum procedure (vacuum chambers) and temperature changes (vacuum and sealed chambers). The storage period ranged from 0 to 93 days. The table shows the results of three Tukey comparisons of means, one for each dependent variable (air and O$_2$ availability per insect, and O$_2$ availability per mg of live weight).

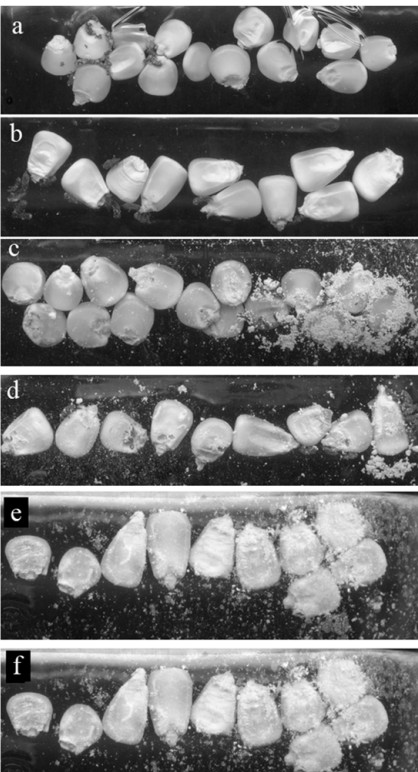

**Figure 1.** Damage to seeds used as sustenance after 93 days to evaluate weevil viability in (**a**,**b**) two vacuum chambers (0.26 ± 0.04 atm (atmosphere)), (**c**,**d**) two hermetically sealed chambers (1 ± 0.08 atm), and (**e**,**f**) two air exchange chambers closed with filter paper (1 atm). Pressure variations are due to the vacuum procedure (vacuum chambers) and temperature changes (vacuum and sealed chambers).

**Table 6.** Damage index inflicted by insects after 93 days of storage in corn grains.

| Chamber | Damage Index, Y | Damage Index, T |
|---|---|---|
| Air exchange chamber (1 atm (atmosphere)) | 4.83 a | 85.57 a |
| Sealed chamber (1 ± 0.08 atm) | 4.17 b | 67.86 b |
| Vacuum chamber (0.26 ± 0.04 atm) | 0.00 c | 4.43 c |
| DMS (Tukey, 0.05) | 0.4999 | 16.26 |

Averages in columns with the same letter do not differ statistically. The damage index was assigned visually. Both the original (Y) and transformed (T) data were used for the analysis; the original data were transformed with an arcsine function [T = arcsine $(Y/5)^{1/2}$]. The table shows the results of two Tukey comparisons of means, one for each dependent variable (original and transformed damage indexes).

## 4. Discussion

### 4.1. Standard Germination

Germination was not significantly reduced after 180 days, as could be expected; furthermore, over a longer period, the vacuum could reduce the oxygen pressure and thus the potential deleterious effect, allowing greater longevity [18,34,35]. Comparatively, the germination of barley seeds, a species considered to have low longevity, in environments with a vacuum, $CO_2$, $N_2$, and air remained approximately the same for a period of 1 to 5 years, but storage in a vacuum allowed significantly higher germination at 15 and 26 years [36].

### 4.2. Weevil Viability

Weevil viability in the air exchange chamber on day 93 corresponded to the life expectancy for insects under stress-free conditions, which is 53 to 126 days [37,38]; meanwhile, insect viability was lower in the sealed and vacuum chambers. The number of insects did not increase, except for the two insects observed in the air exchange chambers (not included in the analysis); this was because their age at the beginning of the test was 60–75 days, which exceeded the most prolific stage, ranging from 40 to 60 days [37].

In studies carried out with 80% $CO_2$ and 20% $N_2$, 100% mortality of *Plodia interpunctella* (Hübner, 1813) pupae was obtained in 12 h [39]. Mortality at these concentrations could rather be attributed to the lack of oxygen than to the toxic effect of $CO_2$ [39,40]. Although these methods are faster in achieving mortality, the use of a vacuum has a comparable speed of action to the most widely used commercial fumigant ($PH_3$), which is 96 h [41,42]; furthermore, under certain conditions, a time of 4 to 192 h was required to achieve 100% mortality in adults of *Oryzaephilus surinamensis* (Linnaeus, 1758) [43] through the use of phosphine ($PH_3$).

Additionally, the lethality of a vacuum is comparable to that of nitrogen, since at a concentration of 98%, it caused the death of 100% of *Callosobruchus maculatus* (Fabricius, 1775) in seven days. In addition, it reduced oviposition and prevented the emergence of adults [44], although total eradication is possible in less time, since Chiappini, Molinari, and Cravedi [45] determined that the time necessary for the total elimination of *Tribolium confusum* Jaquelin Du Val, 1868 insects varied from 1 to 7 days depending on the concentration of $O_2$ and $N_2$, as well as the temperature. In *Callosobruchus maculatus* and *Acanthoscelides obtectus* (Say, 1831), 100% $N_2$ eliminated all adult insects, eggs, larvae, and pupae in 5 days [46].

On the other hand, the use of ozone at 25 ppm caused 100% mortality in two days [47], and although it is effective in the control of *Sitophilus oryzae* A.Hustache, 1930 and *Sitophilus zeamais* Motschulsky & V.de, 1855, as well as other warehouse pests [48], it presents a high risk of corrosion to facilities and structures based on metals (with some exceptions), and even rubbers and other organic materials may be damaged. Furthermore, ozone has a potential irritant effect on humans and must be supplied continuously because its half-life is 20 to 50 min [49]; nevertheless, applications of 120 min have been studied, and 100% mortality of *Sitophilus granarius* (C. Linnaeus, 1758) has been obtained seven days after ozonation [50]. At present, some companies carry out the disinfestation of silos using ozone, thus eliminating insects and fungi in a safe way.

Vacuums have been used for the fumigation of various products since about 1910, and by 1959, they were already being used commercially to improve the penetration of chemicals [51]. After vacuum application (around 0.1 atm (atmospheres)), the fumigant and air are allowed access to the facilities. Some fumigants require ambient humidity, and thus their combined use with a vacuum is ineffective or requires a higher dose than at atmospheric pressure, although recirculation of the air inside the chamber increases the effectiveness of the fumigant. Brown [51] mentioned that the use of a vacuum with the usual time and pressure is not enough to eradicate certain types of insects, such as the grain weevil, but in his reference, the duration of the tests was shorter than the 4 days proven to be effective in the present study. The vacuum applied in the present study ($0.26 \pm 0.02$ atm)

was lower than that recommended by the USDA [52] for facilities dedicated to vacuum-assisted fumigation (0.095 to 0.087 atm); however, weevil eradication was effective on the 4th day in our study, which differs from the time targeted by the USDA [52], which is about 1 to 4 h [51] because a time shorter than 4 days is desirable under some circumstances, such as quarantine stations. It is important to note that some fumigants, such as aluminum phosphide or phosphide of magnesium, are not recommended to be applied with the aid of a vacuum [41] because the lower humidity content of the environment might restrain the production of phosphine ($PH_3$); however, phosphine itself can be applied with the aid of specialized equipment.

The disinfestation of insect pests has also been studied by means of a vacuum combined with monoterpenoids, temperature, and relative humidity. The highest mortality was obtained when the vacuum was associated with higher storage temperatures, 30–35 °C vs. 5–25 °C [7], which were also associated with a shorter period for causing the death of insects.

The vacuum condition can be maintained, according to the results of the present study, for a period of up to at least 180 days without negatively affecting the germination of the seeds, or putting the safety of the operators at risk by poisoning, and although the effect on other stages of the weevil was not observed in the present study, it is considered that during the storage period, it is possible to keep the presence of adults under control, thereby achieving safe storage.

### 4.3. Air and Oxygen Availability per Insect

The air renewal rate in the chamber with air exchange allows the air to be renewed within 6.25 days. It can therefore be stated that the death of the insects in these chambers was due to their natural longevity and not to a toxic effect or a lack of oxygen. The availability of oxygen per unit mass of the insects in the vacuum, sealed, and gas exchange chambers was higher than the oxygen consumption of various species such as *Tribolium castaneum* (Herbst, 1797), *Tenebrio molitor* Linnaeus, 1758, *Cotinis mutabilis* (Gory & Percheron, 1833), and *Dynastes granti*, Horn, 1870, which can consume 0.047, 0.045, 0.062, and 0.016 µmol $h^{-1}$ $mg^{-1}$ according to Lease and Klok [24]. This allows us to deduce that, in addition to the oxygen deficiency and the toxic effect produced by the $CO_2$ exhaled by the insects, the vacuum pressure contributed to the decrease in their viability, probably by limiting breathing or by causing dehydration.

### 4.4. Grain Damage

The results clearly show that vacuum chambers can prevent any damage to the grains and seeds, even if the insects survive for a few days; meanwhile, air exchange or sealed chambers are not suitable for preventing damage inflicted by insects.

### 4.5. General Discussion

According to our results, a partial vacuum environment can be used to safely store seeds free of synthetic insecticides without negative effects on seed germination; in addition, it can prevent adult insect survival. Further research on preventing the survival and development of eggs, larvae, and pupae might be required. Our findings may find practical application in the vacuum storage of germplasms, with the possibility of dispensing in a cold room since the experiment was carried out at room temperature (24–26 °C); however, further research with a longer storage period is necessary. Our research may also find practical application in the storage and transfer of grains in vacuum chambers, with the possibility of eliminating or reducing insecticide applications. The results obtained could contribute to addressing current challenges by reducing the number of applications of synthetic substances in seeds and stored grains and their consequent risks to the applicator, the consumer, wildlife, and the environment, as well as by reducing energy use in the application of cold for the storage of seeds and germplasms. The technology used for vacuum storage is relatively simple and economical.

## 5. Conclusions

The vacuum pressure did not negatively affect the germination of the seeds over a period of 180 days. Under these conditions, all the adult insects are eliminated over a period of four days, and physical damage to the grains or seeds is totally avoided, representing a possible solution for the safe storage of seeds and grains. On the other hand, hermetic packaging or gas exchange packaging is not safe for storage (under the evaluated conditions), as it allows for insect activity and physical damage to the grains or seeds. The technology used for vacuum storage is relatively simple and economical under the evaluated conditions. According to the results, the partial vacuum environment can be used to safely store seeds and grains free of any toxic products.

**Author Contributions:** Conceptualization, methodology, and writing—original draft preparation A.M.-R.; formal analysis, A.M.-R and E.G.-E.; conceptualization, review, and editing, E.G.-E., M.E.V.-B., E.C.-C. and M.S.-V. All authors have read and agreed to the published version of the manuscript.

**Funding:** This research was funded by Universidad Autónoma Agraria Antonio Narro, Proyecto interno 38111-425105001-2211, Atmósfera modificada por presión y nitrógeno para el control de *Sitophilus zeamais* y *S. oryzae* en semilla de maíz.

**Institutional Review Board Statement:** Not applicable.

**Data Availability Statement:** All the data incorporated in the manuscript.

**Conflicts of Interest:** The authors declare that the research was conducted in the absence of any commercial or financial relationships that could be construed as a potential conflict of interest.

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
