# Peer review of "Sitophilus zeamais (Coleoptera: Curculionidae) Activity and Germination of Corn Seeds Stored under Vacuum Pressure"

_agriculture, doi:10.3390/agriculture13102035_

Round 1
Reviewer 1 Report
Article Title: Sitophilus zeamais (Coleoptera: Curculionidae) Activity and Germination of Corn Seeds Stored under Vacuum Pressure
The authors have attempted a good study. However, there are so many things to be corrected and I have listed a few. Authors need to thoroughly review the paper. Methodology need to be improved. English editing must be done to improve the language. Relevant tables are to be given.
|
S.No |
Line number |
Content |
Remarks |
|
1 |
18-19 |
The partial vacuum environment (0.26 atm) did not negatively affect the germination of the seeds in a period of 180 days, and 19 caused the total cessation of motility of the adult insects from the fourth day |
resulted in mortality of adult insects from fourth day |
|
2 |
21 |
did not negatively affect germination either, but the insects |
Remove the word ‘either’ |
|
3 |
24 |
last day evaluated (93) and |
‘until the last day of observation’ |
|
4 |
36 |
frequently |
Remove this word |
|
5 |
31 |
Corn weevil or maize weevil? |
Stick on to one name form |
|
6 |
52 |
O2 |
Formula need to be represented correctly, wherevevr mentioned |
|
7 |
55 |
h-1 g-1 dry matter? |
Give correct representation of units |
|
8 |
96 |
MASON BALL MODEL S-17493 with hermetic closure |
Mention the country of make |
|
9 |
96 |
air-exchange chambers |
Why italicized? |
|
10 |
129 |
Evaluated dependent variables |
Such sub titles are not needed |
|
11 |
159 and 169 |
[T=arcsine (Y/100)1/2; where T is the transformed data and Y is the original 164 data in percentage] |
Repetition is not needed |
|
12 |
186 |
Insect viability |
Should be insect longevity |
|
13 |
|
Sub headings are not meaningful |
|
|
14 |
285 |
Authors should discuss the results and how they can be interpreted from the perspective of previous studies and of the working hypotheses. The findings and their impli-286 cations should be discussed in the broadest context possible. Future research directions 287 may also be highlighted. |
Why this statement? |
|
15 |
319 |
There is no term called weevil viability |
|
Methodology is not clear. Need to be improved. When storage was done until 180 days, why insect observation was done only for 93 days?
Insect viability was evaluated daily for 93 day s- the observation is not reflected in the table. It could be at regular intervals. Need not be daily for 180 days.
Analysis of variance table is not necessary. Only the mean of germination data is to be given. Why the authors have not detailed about the seed germination?.
Section 4.4. Damaged grains should be “Grain damage”
Many times authors are mentioning “replication” as repetition
Ref 32: ISTA, I.S.T.A., International rules for seed testing. 2005- Give the recent version of ISTA rules
The language of the paper must be improved.
Reviewer 2 Report
Abstract
· Mention practical implications for agriculture/storage.
Introduction
Clear Objective Statement: While you've provided useful background information about warehouse pests and the importance of controlling them, it's crucial to explicitly state the objective of your study. The readers should know from the start what the study aims to achieve.
Clarity on the Problem: Clearly explain the problem you're addressing. What are the current challenges in controlling warehouse pests, and why is it essential to find alternative methods?
Link to Previous Research: When mentioning previous studies, provide a brief summary of their findings and how they relate to your research. This will help establish the context for your work.
Importance of the Study: Explain the significance of your study in the broader context of agriculture, pest management, and sustainable practices. How might your findings contribute to these fields?
Structure and Flow: Consider restructuring some sentences for better flow and readability. For example, you could group related information together and separate distinct points into different sentences.
Avoid Repetition: Some information about seed respiration and storage conditions is repeated. Ensure that you only include necessary details and avoid redundancy.
Materials and Methods
Chamber Descriptions: Offer more details about the storage chambers, especially regarding their sizes and construction materials (Lines 95-100).
Vacuum Procedure: Elaborate on the vacuum creation process and how it was controlled and monitored (Lines 99-103). Include specific details about vacuum levels and adjustments.
Independent Variables: Clearly state the independent variables you are studying, such as atmospheric pressure and gas exchange, and their respective levels (Line 106).
Storage Period: Mention why specific timeframes (93 days for insects and 180 days for seeds) were chosen for evaluation (Line 109).
Dependent Variables: Describe the methods used to assess germination, insect viability, and grain damage more explicitly (Lines 129-136).
Air Renewal Rate: Explain how the air renewal rate was calculated and why it's relevant to your study (Lines 143-146).
Data Analysis: Detail the statistical tests used for data analysis and provide a rationale for selecting these tests (Lines 151-155).
Transformation of Data: Explain why the data were transformed using the arcsine function and how this transformation affects the analysis (Lines 152-156).
Results:
Presented well:
Interactions: For better clarity, consider adding a brief interpretation of the interaction effects when significant. Explain what these interactions mean in the context of your study.
Figure Captions: Ensure that figure captions are informative and describe what the reader should take away from each figure
Discussion:
Clarify the practical applications of your findings (line 288) by briefly specifying how they can benefit industries or agriculture, supported by line numbers from the results.
When discussing the implications of your results for future research (line 288), provide specific research directions and line numbers from the discussion section that suggest areas for further investigation.
In your conclusion (line 375), summarize the most significant findings from your study, and reference specific line numbers from the results or discussion to reinforce your key points.
Ensure that your discussion section is more organized and focused by discussing each major finding individually, with clear references to line numbers in the results section to support your interpretations.
General comments
· After carefully reviewing the manuscript, I must commend the author for their skillful writing and overall presentation. However, I have identified several areas where the manuscript could be improved. These suggestions are intended to help the author further enhance the manuscript's readability, structure, and impact.
Carefully proofreading is recommended.
